# Evaluation of the Immune Response to Transport Stress in the Aosta Valley Breed

**DOI:** 10.3390/vetsci10050351

**Published:** 2023-05-14

**Authors:** Giulia Pagliasso, Martina Moriconi, Francesca Fusi, Nicoletta Vitale, Mario Vevey, Alessandro Dondo, Elisabetta Razzuoli, Stefania Bergagna

**Affiliations:** 1Instituto Zooprofilattico Sperimentale del Piemonte, Liguria e Valle d’Aosta, Via Bologna 148, 10154 Torino, Italy; 2Azienda Sanitaria Locale di Ciriè, Chivasso e Ivrea, Via Cavour 29, 10073 Ciriè, Italy; 3Italian National Reference Centre for Animal Welfare (CReNBA), Istituto Zooprofilattico Sperimentale della Lombardia e dell’Emilia Romagna “Bruno Ubertini” (IZSLER), Via A. Bianchi 9, 25124 Brescia, Italy; 4Associazione Nazionale Bovini di Razza Valdostana, Fraz. Favret, 5, 11020 Gressan, Italy

**Keywords:** transport, stress, immune response

## Abstract

**Simple Summary:**

From an animal welfare point of view, transportation is regarded as an acute stressor often disrupting the internal homeostasis of animals. Transportation provides a variety of physical and mental stressors related to pre-transportation rearing conditions, handling, loading, voyage conditions, unloading, and post-transportation housing conditions, implying adaptation to a new environment. The extent and the duration of stressors play an important role in shaping and modulating homeostatic control responses. The present study, carried out on 45 young bulls, aimed to determine whether short-term transportation could alter selected hematochemical and immunological parameters. The results obtained showed low-grade and transient alterations of the analysed parameters, which did not seem to compromise the bulls’ well-being.

**Abstract:**

Transportation is a recurring event in a farm animal’s life, and it is considered one of the main stressors with possible negative repercussions for both the health and welfare of farm animals. The objective of the present study was to examine the effect of transportation on some blood variables of 45 young bulls moved from their original farms to a livestock collection centre. Transportation took no more than 8 h and was carried out between January and March 2021. Blood samples were taken before transportation (T0), upon arrival at the collection centre (T1), and 7 days after arrival (T2). Samples were processed for blood cell count, clinical chemistry analyses, serum protein electrophoresis, and the evaluation of innate immunity parameters. The results showed a typical stress leukogram with neutrophilia and changes in the neutrophil:lymphocyte ratio. No significant alterations were observed in either serum proteins or pro-inflammatory cytokines. Significant, albeit transient, alterations were observed in some clinical chemistry parameters after transportation, which could be accounted for by stressful conditions such as the transportation itself and handling and mixing with other animals. Our results indicated that the adopted transportation conditions only slightly affected the blood variables under study with no significant impact on animal welfare.

## 1. Introduction

Transportation from farm to farm or from farm to slaughterhouse is an almost unavoidable event in a farm animal’s life. Handling and transportation represent critical phases as they cause stress with negative consequences for both animal health and welfare [1,2]. Even with the current, greatly improved methods of shipping, transportation is still considered one of the most stressful events experienced by cattle during their lives [2]. That is the reason why the European Community has addressed the issue of transportation through the adoption of specific regulations aimed at protecting animal welfare [3].

Transportation involves several potential stressors including handling, loading, unloading, mixing with other animals, trailer accelerations and vibrations, noise, changes in temperature and humidity, inadequate ventilation, frequent deprivation of food and water, social regrouping, and penning in a new and unfamiliar environment [4,5,6]. Importantly, animals can be exposed not only to physical stressors such as hunger, thirst, fatigue, injury, or thermal extremes, but also to psychological stressors such as fear due to restraint, handling, and novelty [7]. Consequently, considering all these factors, an animal’s response to transportation stress is quite complex and there is little information on the contribution of each stressor. It is nearly impossible to design a study to identify the influence of all the factors cited above [2].

In the veterinary context, the term “stress” can be defined as “an abnormal or extreme adjustment in the physiology of an animal to cope with adverse effects of its environment and management” [8]. The stress response consists of a set of physiological mechanisms aimed at returning to homoeostasis [9]. One response to acute stressors is the activation of the hypothalamic-pituitary-adrenal (HPA) axis, resulting in elevated corticotropin releasing hormone (CHR), which stimulates the anterior pituitary gland to release adrenocorticotropic hormone (ACTH) and other peptides. Increased ACTH causes the release of glucocorticoids and catecholamines in the serum of stressed farm animals [1,10]. Circulating cortisol as an indicator of the HPA activation is the most commonly utilised measurement, and increases have been observed in nearly all transportation studies of cattle as compared with the pre-transportation concentrations or those obtained from non-transported counterparts [9].

It has generally been thought that activation of the HPA axis following stress is responsible for the reduction or suppression of immunocompetence [2]. Therefore, one of the concerns regarding the transportation of cattle is that any resulting stress may render the animals more susceptible to disease [5,11]. In fact, transportation stress is considered to be significantly associated with the bovine respiratory disease known as shipping fever [12,13]. Moreover, young calves are more vulnerable than mature cattle to transportation stress because of an immature immune system and a lack of exposure to new environments [2]. Nevertheless, it has not been widely accepted that environmental stressors directly alter the immune defence mechanisms of the host [14].

Considering that transportation as a stressful event is likely to affect the immune system, the aim of this study was to investigate the evolution pattern of haematological and haematochemical parameters and some specific immunological markers of young bulls during short journeys from their original farms to a genetic centre.

## 2. Materials and Methods

### 2.1. Study Design

The Aosta Red Pied (Valdostana Pezzata Rossa (VRP)) and the Aosta Black Pied (Valdostana Pezzata Nera (VBP)) cattle have been raised for a long time, and have been selected to be farmed in a mountain environment and be adapted to the hostile conditions of alpine territories. They are dual-purpose cattle breeds, milk and meat, farmed in semi-intensive systems characterised by summer pasture at very high altitude and winter housing in farms [15]. Under the supervision of the Associazione Nazionale Allevatori Bovini di Razza Valdostana (A.N.A.Bo.Ra.Va.) (National Association of Breeders of the Valdostana Breed) [16], the young bulls entered the control station at 45–60 days of age; next, weaning began at 130/135 days and was over at 150 days. At this point, the animals were admitted to performance tests up to 11 months of age when they were brought to the breeding station.

A total of 45 bulls, specifically 28 VRP and 17 VBP, coming from 40 different farms located in the Valle d’Aosta territory were included in the study. All subjects were registered in the studbook of the Aosta Valley breed which represents the tool for the selective improvement of the breed.

The animals were 60.7 ± 8.5 days old and their weight was 84.2 ± 12.1 kg. Transportation was carried out between January and March 2021. A total of 35 calves came from 35 different farms, while in the remaining 5 farms 2 calves were loaded from each. The animals were transported from a specific farm straight to the genetic centre during the course of a single day. No calves from different herds were loaded together. A commercial truck with a maximum capacity of 12 calves was used, equipped with dividers, shaping the space into smaller sections to prevent falls and crushing from acceleration; both natural and mechanical ventilation was adopted, and the animals were provided with an adequate amount of chopped straw litter. Transportation started immediately after loading the bulls and took an average of 188.4 ± 123.8 min. Upon arrival, antiparasitic treatment (Virbamec, Virbac) was applied to all bulls at the same time on 11 March, while a combined vaccination against Respiratory Syncitial and Parainfluenza 3 viruses (Rispoval Intranasal Rs+PI3, Zoetis) was carried out at different times after the arrival of the animals at the centre.

Sampling was carried out three times: in the original farm before transportation (T0), upon arrival at the control station (T1), and 7 days from arrival (T2). Blood samples were taken from the jugular vein of the bulls using vacutainer tubes with K3EDTA anticoagulant and without anticoagulant. Whole blood samples from K3EDTA tubes were immediately processed for blood counts, while tubes without anticoagulant were centrifuged at 3500 rpm for 15 min at 20 °C. The serum was separated from the clot and stored in 1.5 mL aliquots at a temperature of −80 °C until the analysis was performed. The serum aliquots were subjected to clinical chemistry analysis, to protein electrophoresis, and to the evaluation of innate immunity parameters. The experimental study was approved by the Ethics Committee on Animal Use of the Istituto Zooprofilattico Sperimentale of Piemonte, Liguria and Valle d’Aosta, protocol n° 3272.

### 2.2. Complete Blood Count (CBC) Analysis

Upon arrival at the laboratory, fresh whole blood samples containing K3EDTA anticoagulant were immediately subjected to analysis for the determination of the blood count using appropriate instrumentation (MS4 Instrument), according to the instructions provided by the manufacturer.

Blood counts included: erythrocytes (RBC), haemoglobin (HB), hematocrit (HCT), mean cell volume (MCV), mean corpuscular haemoglobin (MCH) and mean corpuscular haemoglobin concentration (MCHC), leukocytes (WBC) with leukocyte formula (lymphocytes (LINF), monocytes (MON), neutrophils (NEU), eosinophils (EO), and basophils (BA)), and platelets (PLT). The reference values used for the analysis were those set in the automatic analyser through a specific validation procedure.

### 2.3. Clinical Chemistry Analysis

The serum samples previously stored at −80 °C were brought to room temperature and then mixed for a few seconds on Vortex, carefully avoiding the formation of foam. The serum aliquots were subjected to clinical chemistry analysis using an automated photometer (ILab Aries Chemical Analyzer—Instrumentation Laboratory) following the manufacturer’s instructions. The analysed parameters were: albumin (ALB), alkaline phosphatase (ALP), alanine aminotransferase (ALT), aspartate transaminase (AST), total bilirubin (BILT), calcium (CA), cholesterol (CHOL), creatinine (CREAE), iron (FE), gamma-glutamyl transferase (GGT), chlorine (CL), potassium (K), sodium (NA), magnesium (MG), phosphorus (PHOS), total protein (TP), triglycerides (TRIG), and urea (UREA). The reference values used for the analysis were obtained from the bibliography [17].

### 2.4. Serum Protein Electrophoresis

The serum previously stored in the freezer at −80 °C was used for the evaluation of the serum proteins. The analysis, which consisted of a zonal electrophoresis of the serum proteins, was carried out by separating the fractions based on their net electric charge in an alkaline medium (pH 9.2) on a 0.8% agarose gel, stained with a Amidoscharwz solution. The test was performed using a semi-automatic multiparametric biochemical analyser for electrophoresis (HYDRASYS LC SEBIA, HYGRAGEL PROTEIN 15/30 kit and densitometer). The serum proteins following electrophoresis were separated into 4 different types of fractions (protidogram): albumin (ALB), α-Globulins (α-G), β-Globulins (β-G), and γ-Globulins (γ-G).

### 2.5. Immunoenzymatic Analysis

The analysis to quantify the plasma levels of interleukins and the acute phase proteins (APPs) was carried out using commercial kits according to the indications and standard procedures provided by the manufacturer. In particular, the following ELISA kits from the Bioassay Technology Laboratory (Shanghai, China) were used:-“Bovine Interleukin 6 ELISA Kit” (Standard curve range: 20–6000 ng/L; Sensitivity: 10.5 ng/L),-“Bovine Tumor Necrosis Factor Alpha ELISA Kit” (Standard curve range: 10–3000 ng/L; Sensitivity: 5.56 ng/L),-“Bovine Serum Amyloid A ELISA Kit” (Standard curve range: 0.4–40 µg/mL; Sensitivity: 0.054 µg/mL).-“Bovine Haptoglobin ELISA Kit” (Standard curve range: 3–900 µg/mL; Sensitivity: 1.69 µg/mL).

### 2.6. Statistical Analysis

The sample size was calculated for an ANOVA study design with repeated measures within factors. The following parameters were considered: alfa error of 0.05, effect size of 0.25, power of 0.95, correlation among repeated measures of 0.5, and 3 measurements (T0, T1, T3).

## 3. Results

The blood counts did not detect significant changes in the parameters at the various sampling times. In particular, HCT did not increase after transportation and the HB value was always higher than the minimum limit set by law, equal to 7.25 g/dL [18]. The animals in the present study showed typical stress leukograms with neutrophilia and changes in the neutrophil:lymphocyte ratio between T0 and T1 and a simultaneous reduction in lymphocytes, even if not statistically significant (Table 1).

The analysed clinical chemistry parameters showed some relevant alterations regarding animal metabolism (Table 2). Upon arrival at the control station (T1), there was an increase in AST and CL parameters, while TRIG decreased. Seven days after arrival (T2), the renal parameters CREA and UREA increased, while ALP, CHOL and TRIG decreased.

No significant differences were identified at the different sampling times for any parameter analysed for serum protein electrophoresis (Table 3).

Quantified acute phase proteins and interleukins did not reveal significant alterations over time (Figure 1).

## 4. Discussion

The innate immune system is able to respond to both infectious stressors and non-infectious stressors [19]. The first effect of stress in animals is a transient increase of glucocorticoids and catecholamines that can modulate a variety of biological effects also affecting immune functions. Therefore, there is a need to effectively control actions preventing an excessive release of pro-inflammatory cytokines (e.g., IL-6, IFN-γ and TNF-α) and other components leading to severe inflammation and tissue damage [20]. In this respect, exposure to stressors such as transportation, mixing, and weaning were shown to lead to increased blood concentrations of APPs including SAA and APTO [21].

During road transportation, the truck environment and other parameters such as truck design, stocking density, driving style, road quality, ventilation, and ambient temperature can be conducive to tissue damage, discomfort, and added stress [22]. This combination of acute, non-infectious stressors is responsible for increased cortisol, altered energy and protein metabolism, changes in appetite and growth rate, and a challenged immune system [5,23].

In this study, the transportation conditions were not particularly severe, as the animals experienced short journeys (less than 8 h) carried out in the winter period (between January and March) at low ambient temperatures. Furthermore, calves from the same farm were transported singly or at most in pairs; consequently, there was no mixing on the conveyance.

However, the analyses showed variations in some parameters, presumably due to the combined stress of transportation and new farming environment. As also observed by Lomborg et al. (2008) [21], the results obtained in this study showed a typical stress leukogram with neutrophilia and changes in the neutrophil:lymphocyte ratio between T0 and T1 and a simultaneous reduction in lymphocytes, even if not statistically significant. Specifically, the increase of the neutrophil:lymphocyte ratio observed at T1 (>1) could be considered a stress indicator [24]. These alterations were not caused by hemoconcentration as hematocrit did not change over time, but they could be the result of stressful factors, specifically transportation, rather than dehydration or pathological conditions. Furthermore, these values returned to physiological levels 7 days after arrival in the genetic centre.

Regarding the clinical chemistry parameters, a significant increase of AST was observed right after the arrival of the animals at the genetic centre that could be explained by glucocorticoids-induced muscle catabolism [25], which was confirmed by the significant increase of UREA and CREA 7 days after arrival. At the same time, the significant decrease of cholesterol and triglycerides could indicate a change in the animals’ metabolism. Additionally, the significant decrease of ALP at T2 could be explained by the different environmental and feeding conditions of the animals compared to the origin. Indeed, previous studies showed lower serum alkaline phosphatase content in zinc-deficient calves [26], while Halse et al. (1948) [27] reported that the phosphatase activities in the blood serum of cows tended to be low during hypomagnesaemia. Stress could be responsible also for respiratory alkalosis as a consequence of an increase in respiratory rate that was observed during heat stress [28]. While respiratory frequency has not been evaluated in this study, the significant increase in CL observed right after arrival at the genetic centre could be explained by respiratory alkalosis as a result of other stressful conditions such as handling or mixing with other animals.

No significant alterations were observed for albumin and the other serum proteins. The study of Mormede et al. (1982) [29] showed that the serum albumin and protides of calves increased on arrival at the farm but were significantly higher in animals subjected to a long journey than in animals subjected to a short journey. Serum immunoglobulin levels, instead, did not change upon transportation. Another study showed a seasonal effect for albumin and alpha-globulin serum fractions, as opposed to beta and gamma-globulins [19].

As for APPs, no significant alterations were observed between the various time points investigated. Regarding APTO, previous works demonstrated a serum concentration increase in calves after long transportation (about 1000 km), as well as a 10-fold increase in SAA [30]. However, the most significant aspect of the APP response was transporting cattle over long distances without stops for food, water, and rest [31].

No significant alterations were observed for the pro-inflammatory cytokines investigated (IL-6 and TNF-*α*). IL-6 was the major mediator underlying the hepatocytic secretion of most APPs [25], and its non-significant increase after transportation could partly account for the lack of an APP response. TNF-α causes muscle catabolism that is also mediated by glucocorticoids, as well as by glucagon, inducing hyperglycemia and amino acid uptake by the liver [25]. TNF-α in serum alone has been found to increase after transportation [30], while another study reported an increase in pro-inflammatory cytokines (IL-6 and TNF-α) over 15–30 days after arrival in a novel environment [19].

As observed by Cafazzo et al. [1], in this study as well, the effect of a short road journey only slightly affected the response of animals. This showed a transient change in the leukogram not associated with increased levels of pro-inflammatory cytochines, and also a significant alteration of fat and protein metabolism probably due to transportation and adaptation to the new environment. Different results from those observed in our study have been reported in cases of long-term transport. Li et al. (2019) observed a significant increase in IL-6 and TNF-α in cattle transported for 350 km in 6 h [23]; another study showed a decrease in INF-γ, lymphocyte, and body weight and a significant increase in neutrophils, eosinophils, packed cell volume, red blood cell numbers, and haemoglobin in bulls transported for 12h by road [32]; additionally, an increase in total protein albumin, urea, and haptoglobin plasma concentrations has been reported following transportation for 18 h [33]. It is to be noted that the evaluation of the effects of transportation on different cattle breeds showed that the factor of breed significantly affected the concentration of TNF-α and IL-6, meaning that different breeds have a different inflammatory reaction during transportation stress [23]. The transportation stress response may be minimised by careful handling, good facility design, appropriate storage densities, good driving techniques, and short journey time [34].

## 5. Conclusions

In conclusion, the observed blood variations were considered compatible with the young bulls’ welfare. Further research is required to fully understand the relationships between transportation stress, immune system status, and the risk of post-transportation disease development.

## Figures and Tables

**Figure 1 vetsci-10-00351-f001:**
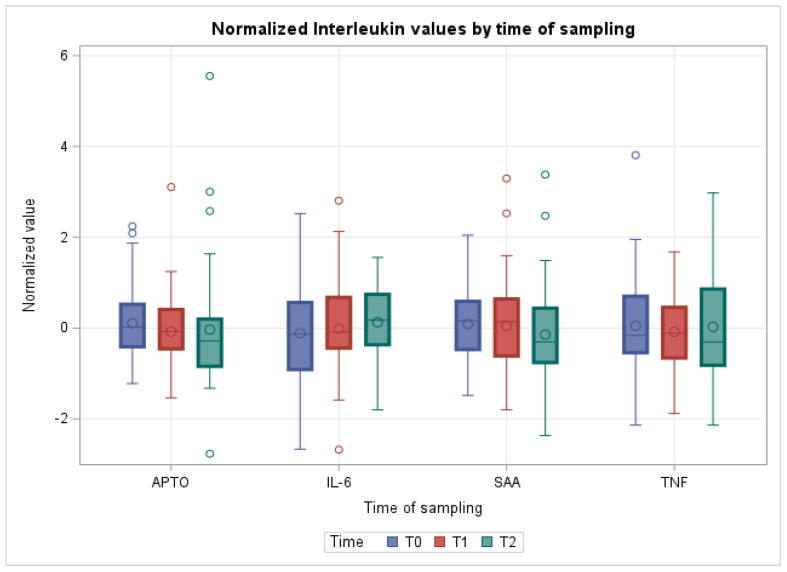
Boxplots showing cytokines quantification in the different sampling times. Data are normalized. APTO: haptoglobin; IL-6: interleukin-6; SAA: Serum Amyloid; TNF: tumor necrosis factor α.

**Table 1 vetsci-10-00351-t001:** Means of blood parameters at the different sampling times. The results are expressed as mean ± SD.

	T0	T1	T2
BA %	0.44 ± 0.26	0.44 ± 0.26	0.39 ± 0.26
EO %	1.32 ± 3.82	0.51 ± 1.12	0.33 ± 0.78
WBC (m/mm^3^)	10 ± 2.83	** 11.7 ± 2.5	9.77 ± 2.69
RBC (m/mm^3^)	10.1 ± 1.28	10.2 ± 1.18	9.73 ± 1.24
HB (g/dL)	12.8 ± 2.21	12.8 ± 2.01	12 ± 2
HCT %	34.3 ± 6.42	34.2 ± 6.05	32.8 ± 6.12
LINF %	51.7 ± 11.7	45.3 ± 13.2	51.3 ± 12.4
MCH (pg)	12.6 ± 1.33	12.5 ± 1.07	12.3 ± 0.93
MCHC (g/dL)	37.3 ± 3.17	37.6 ± 3.01	36.8 ± 2.77
MCV (fL)	33.9 ± 3.53	33.6 ± 3.39	33.6 ± 3.45
MON %	6.68 ± 2	6.4 ± 3.36	6.85 ± 2.51
MPV (fL)	10 ± 0.75	9.96 ± 0.77	9.77 ± 0.85
NEU %	39.9 ± 13.2	* 47.3 ± 15.2	41.1 ± 13.9
PCT %	0.54 ± 0.35	0.59 ± 0.39	0.72 ± 1.16
PDW	8.99 ± 2	8.79 ± 1.96	8.62 ± 1.89
PLT (m/mm^3^)	518 ± 310	578 ± 347	549 ± 301
RDW	17.3 ± 2.3	17.2 ± 2.11	16.9 ± 2.29
RRg	3.45 ± 8.64	4.29 ± 9.52	1.65 ± 4.05

* Indicates a significant difference assessed using the ANOVA test with respect to T0 (*p* < 0.05). ** Indicates a significant difference assessed using the ANOVA test with respect to T0 (*p* < 0.01).

**Table 2 vetsci-10-00351-t002:** Means of blood clinical chemistry values at the different sampling times. The results are expressed as mean ± SD.

	T0	T1	T2
ALP (UI/L)	568 ± 279	575 ± 271	*** 246 ± 124
ALT (UI/L)	17.3 ± 4.83	17.2 ± 5.45	15.7 ± 5.41
AST (UI/L)	76.3 ± 19.1	* 87 ± 24.5	73 ± 18.3
BILT (mg/dL)	0.2 ± 0.1	0.25 ± 0.13	0.23 ± 0.12
CA (mg/dL)	9.43 ± 3.45	9.45 ± 3.56	8.06 ± 2.82
CHOL (mg/dL)	125 ± 47	127 ± 49.4	** 91.1 ± 40.2
CL (mEq/L)	99.8 ± 7.73	* 104 ± 9.59	99.3 ± 4.26
CREA (mg/dL)	0.92 ± 0.34	0.95 ± 0.34	** 1.12 ± 0.24
FE (µg/dL)	125 ± 95.7	120 ± 86.9	86.2 ± 57.4
GGT (UI/L)	23.1 ± 6.13	22.7 ± 7.16	19.9 ± 7.32
K (mg/dL)	9.31 ± 5.74	8.88 ± 1.91	12.3 ± 17.6
MG (mg/dL)	1.77 ± 0.65	1.66 ± 0.67	1.87 ± 1.03
NA (mg/dL)	143 ± 8.68	147 ± 10.7	140 ± 4.84
PHOS (mg/dL)	8.25 ± 3.32	7.89 ± 3.15	7.61 ± 3.14
TP (g/dL)	5.14 ± 0.78	5.32 ± 0.81	5.04 ± 0.64
TRIG (mg/dL)	29 ± 17.5	** 20.1 ± 13.3	*** 13.4 ± 6.75
UREA (mg/dL)	13.6 ± 5.86	13.2 ± 5.75	** 19.3 ± 10

* Indicates a significant difference assessed using the ANOVA test with respect to T0 (*p* < 0.05). ** Indicates a significant difference assessed using the ANOVA test with respect to T0 (*p* < 0.01). *** Indicates a significant difference assessed using the ANOVA test with respect to T0 (*p* < 0.001).

**Table 3 vetsci-10-00351-t003:** Means of electrophoresis values at the different sampling times. The results are expressed as mean ± SD.

	T0	T1	T2
ALB (g/dL)	2.57 ± 0.47	2.67 ± 0.51	2.49 ± 0.4
α-G (g/dL)	1.02 ± 0.16	1.03 ± 0.16	1.03 ± 0.14
β-G (g/dL)	0.73 ± 0.15	0.76 ± 0.2	0.71 ± 0.14
γ-G (g/dL)	0.84 ± 0.25	0.85 ± 0.26	0.82 ± 0.23

## Data Availability

The data presented in this study are available on request from the corresponding author.

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
