# Peer review of "Evaluation of the Immune Response to Transport Stress in the Aosta Valley Breed"

_vetsci, 2023, doi:10.3390/vetsci10050351_

Round 1
Reviewer 1 Report
The manuscript entitled Evaluation of the immune response to transport stress in the Aosta Valley breed by Pagliasso et al. concerns the study of the effect of short journeys on blood values in a group of calves transported to a genetic center. The analysis was performed on the same animals at three different times (T0, T1, and T2). Some differences were observed in some clinical chemistry parameters after the transport, which the authors reported most likely due to the transport itself.
I reckon that the manuscript must be improved especially in the material and methods section. Moreover, the conclusions section is missing.
Here the authors can find some points that in my opinion require attention:
Introduction:
L44: Please check this sentence (event in an a farm animal's life).
L49: Add legislative references at the end of the sentence.
L57: ‘which obviously underlies the lack of crucial information regarding transport stress’: it is not clear to me the reason why there is an obvious lack of crucial information concerning transport stress, the authors should clarify the meaning of the sentence.
Material and Methods:
The first part of this section is missing the title.
L95-L102: I reckon the authors should include more details concerning the transport, as it is not clear whether the calves were transported during the same days and grouped from different farms (e.g., FARM A + FARM B + FARM C on a single day) or if the animals were transported from a specific farm to the genetic center during the course of a single day (e.g., FARM A on a single day, FARM B on another single day, etc.). Moreover, truck density during transport should be specified, as it is an important factor as also stated by the authors in the Discussion section.
L107-L115: Is blood sampling routinely performed in these cases? If not, should not the authors have approval from an ethics committee? Please provide this information.
Results:
Table 1-2-3: It seems to me that the results are expressed as mean ± SD but it is not specified anywhere, please clarify this aspect.
Discussion:
L254-L255: The authors should further discuss this point, comparing, if possible, results obtained from animals transported for longer times and/or different conditions (e.g., during the summer).
Conclusions:
This section is completely missing.
References:
The list of references is not sequential (References 1 and 2 are repeated)
Author Response
Reviewer 1
1: L44: Please check this sentence (event in an a farm animal'slife).
Authors’ response: Thank you for the tip, it was a typing error.
2: L49: Add legislative references at the end of the sentence.
Authors’ response: Thank you for your clarification. We have added the legislative reference to the European Regulation 1/2005 on the protection of animals during transport and related operations.
3: L57: ‘which obviously underlies the lack of crucial information regarding transport stress’: it is not clear to me the reason why there is an obvious lack of crucial information concerning transport stress, the authors should clarify the meaning of the sentence.
Authors’ response: Thank you for your suggestion. The concept we wanted to express concerned the lack of information on the influence of individual physical and psychological factors on the response to transport stress and not the lack of information on transport stress itself.
4: Material and Methods: The first part of this section is missing the title.
Authors’ response: we added the title “Study design”.
5: L95-L102: I reckon the authors should include more details concerning the transport, as it is not clear whether the calves were transported during the same days and grouped from different farms (e.g., FARM A + FARM B + FARM C on a single day) or if the animals were transported from a specific farm to the genetic center during the course of a single day (e.g., FARM A on a single day, FARM B on another single day, etc.).Moreover, truck density during transport should be specified, as it is an important factor as also stated by the authors in the Discussion section.
Authors’ response: Thank you for your suggestion. The animals were transported from a specific farm directly to the genetic center during the course of a single day. For each trip, 1 calf or a maximum of 2 calves were transported. In fact, 35 calves came from 35 different farms, while 2 calves each were loaded in the other 5 farms. No calves from different herds were grouped and transported together. We do not know the exact dimensions of the means of transport but we do know that it has a maximum capacity of 12 calves. Considering that 1 or 2 calves were transported for each trip, there are certainly no overcrowding problems.
6: L107-L115: Is blood sampling routinely performed in these cases? If not, should not the authors have approval from an ethics committee? Please provide this information.
Authors’ response: Experimental study was carried out with the approval of local Ethical Committee n° 3272
7: Results: Table 1-2-3: It seems to me that the results are expressed as mean ± SD but it is not specified anywhere, please clarify this aspect.
Authors’ response: Thank you for your suggestion. The results are expressed as mean ± SD. We have included this information on the caption of the tables.
8: Discussion: L254-L255: The authors should further discuss this point, comparing, if possible, results obtained from animals transported for longer times and/or different conditions (e.g., during the summer).
Authors’ response: Thank you for your suggestion. We have integrated the discussion.
9: Conclusions: This section is completely missing.
Authors’ response: Thank you for your suggestion. We have inserted the section of conclusion
10: References: The list of references is not sequential (References 1 and 2 are repeated)
Authors’ response: Thank you for the tip, it was a typing error. We have rearranged the references list.

Reviewer 2 Report
Dear authors,
The paper covers an interesting topic. The format is good. In the M&M, I miss crucial information to interpret the data processing, discussion and conclusion.
Line 85: How can you assure that the animals are of the Aosta Valley breed? Were the parent animals registered in a studbook?
Line 101: the transport is not clear to me: Is a journey always from 1 farm to the control centre? Or are calves loaded at different farms and then transported to the control centre. Was there a specific loading density predetermined? Or was the loading density variable? Were the 45 cattle transported separately? Or were several of the 45 test animals transported in some journeys? I think the transport could be described in more detail.
Number of test animals: 45 bulls: how can you assure that this sufficient?
Like you wrote in line 199-201: 'During road transport, the truck environment and other parameters, such as truck design, stocking density, driving style, road quality, ventilation and ambient temperature, can be conducive to tissue damage, discomfort, and added stress'. I miss some of these parameters in your M&M. As a result, I cannot interpret whether the small variation of your results is caused by the small variation of actual results or by the large variation of co-founding factors. For instance, I note that the duration in transport time varies widely.
Author Response
1: Line 85: How can you assure that the animals are of the Aosta Valley breed? Were the parent animals registered in a studbook?
Authors’ response: Thank you for your clarification. All animals included in the study are registered in the studbook of the Aosta Valley breed which represents the tool for the selective improvement of the breed.
2: Line 101: the transport is not clear to me: Is a journey always from 1 farm to the control centre? Or are calves loaded at different farms and then transported to the control centre. Was there a specific loading density predetermined? Or was the loading density variable? Were the 45 cattle transported separately? Or were several of the 45 test animals transported in some journeys? I think the transport could be described in more detail.
Authors’ response: Thank you for your suggestion. All journeys involved the loading of 1 or at most 2 calves from the same herd and the transport directly to the genetic center. In fact, 35 calves came from 35 different farms, while 2 calves each were loaded in the other 5 farms. No calves from different herds were loaded together. The means of transport used has a maximum load of 12 calves, therefore there have been no episodes of overcrowding given that 1 or 2 calves were transported for each trip.
3: Number of test animals: 45 bulls: how can you assure that this sufficient?
Authors’ response:The sample size was calculated for an ANOVA study design with repeated measures within factors. The following parameters were considered: alfa error of 0.05, effect size of 0.25, power of 0.95, correlation among repeated measures of 0.5, 3 measurements (T0, T1, T3). This point was added in Materials and methods section.
4: Like you wrote in line 199-201: 'During road transport, the truck environment and other parameters, such as truck design, stocking density, driving style, road quality, ventilation and ambient temperature, can be conducive to tissue damage, discomfort, and added stress'. I miss some of these parameters in your M&M. As a result, I cannot interpret whether the small variation of your results is caused by the small variation of actual results or by the large variation of co-founding factors. For instance, I note that the duration in transport time varies widely.
Authors’ response: Thank you for your suggestion. The sentence reported here refers to the bibliography and not to our study. In fact, some parameters have not been evaluated (such as the driving style and the road quality). In the following sentences we have highlighted that the transport was of short duration, even if conducted in the winter period (lines 215-217). We also added that there was no mixing of animals during the journey as they traveled individually or in any case they came from the same farm (lines 217-219).

Round 2
Reviewer 1 Report
The authors have addressed all my comments.
In my opinion, the manuscript can be published in its actual form.